

# Semidiurnal solar tide differences between fall and spring transition times in the Northern Hemisphere

J. Federico Conte[1], Jorge L. Chau[1], Fazlul I. Laskar[1], Gunter Stober[1], Hauke Schmidt[2], and Peter Brown[3]

[1]Leibniz Institute of Atmospheric Physics at the University of Rostock, Kühlungsborn, Germany
[2]Max Planck Institute for Meteorology, Hamburg, Germany
[3]Western University, London, Ontario, Canada

*Correspondence to:* J. Federico Conte (conte@iap-kborn.de)

**Abstract.** We present a study of the semidiurnal solar tide (S2) during the fall and spring transition times in the Northern Hemisphere. The tides have been obtained from wind measurements provided by three meteor radars located at: Andenes (69° N, 16° E), Juliusruh (54° N, 13° E) and Tavistock (42° N, 81° W). During the autumn, S2 is characterized by a sudden and pronounced decrease occurring every year and at all height levels. The spring transition also shows a decrease of S2, but

not sudden and that ascends from lower to higher altitudes during an interval of ∼ 15 to 40 days. To assess contributions of different semidiurnal tidal components, we have examined a 20-year free run simulation by the Hamburg Model of the Neutral and Ionized Atmosphere (HAMMONIA). We found that the differences exhibited by the S2 tide between equinox times are mainly due to distinct behaviors of the migrating semidiurnal and the non-migrating westward propagating wave number 1 tidal components (SW2 and SW1, respectively). Specifically, during the fall both, SW2 and SW1 decrease, while during the

spring time SW2 decreases but SW1 remains approximately constant or decreases only slightly. The decrease shown by SW1 during the fall occurs later than that of SW2 and S2, which indicates that the behavior of S2 is mainly driven by the migrating component. Nonetheless, the influence of SW1 is necessary to explain the behavior of S2 during the spring. In addition, a strong shift in the phase of S2 (of SW2 in the simulations) is also observed during the fall. Our meteor radar wind measurements show more gravity wave activity in the autumn than during the spring, which might be indicating that the fall decrease is partly due

to interactions between SW2 and gravity waves.

## 1 Introduction

It is well known that the mesosphere and lower thermosphere (MLT) variability is strongly influenced by a large variety of waves that dynamically interact and couple different regions of the terrestrial atmosphere. Global scale waves include planetary waves (PW), which have periods of ∼ 2-30 days, as well as thermal tides, which have periods that are harmonics of the solar

day (e.g., Rossby, 1939; Forbes, 1984). Gravity waves (GW), on the other hand, are local scale waves characterized by shorter vertical wavelengths and periods of minutes to a few hours (e.g., Fritts and Alexander, 2003). Thermal tides are mainly excited by solar heating of water vapor and ozone, and due to the excitation processes, they have typical periods of one solar day (24 $h$) and its two first harmonics, i.e., 12 $h$ and 8 $h$. When the tides propagate Sun-synchronously, they are identified as migrating.





The non-migrating tides are primarily excited by tropospheric latent heat release and may be westward or eastward propagating (Hagan and Forbes, 2002, 2003).

Given that tides are the dominant waves in the MLT region, they play a significant role in coupling processes by, for example, modifying the propagation conditions for other waves (e.g., Eckermann and Marks, 1996; Smith, 2012). Although there is

considerable tidal variability at different seasons, analyses of the tidal seasonal behavior show that solar tides exhibit significant variations mainly during time periods with strong changes in the mean winds, e.g., during sudden stratospheric warming events (e.g., Charlton and Polvani, 2007; Fuller-Rowell et al., 2010). The spring and fall transitions also show strong changes in the mean winds (e.g., Shepherd et al., 1999; Taylor et al., 2001; Matthias et al., 2015). Hence, one may expect an enhanced and different response of the tides during these time periods. Previous studies have investigated the behavior of solar tides during

equinox times (e.g., Riggin et al., 2003; Pancheva et al., 2009). More recently, Chau et al. (2015) reported a persistent and sudden decrease of the semidiurnal solar tide (S2) in the Northern Hemisphere during the September/October months. However, they did not provide an explanation of this observed sharp decrease of S2. Laskar et al. (2016) speculated that the enhanced S2 amplitudes observed during August/September over Andenes (northern Norway) and Julisuruh (northern Germany) might be due to in-phase interaction between the migrating semidiurnal and the non-migrating westward propagating wave number

1 semidiurnal tidal components, but they could not verify this due to a lack of global datasets. This motivated us to further investigate the differences in the response of the S2 tide between the spring and fall transition times in the Northern Hemisphere using both, observations and model simulations.

The structure of this paper is as follows. In Section 2, we present and describe the tidal features in both, radar measurements and model simulations. Section 3 is used to discuss the comparison of observations with model simulations, in order to explain

the differences seen in the behavior of the semidiurnal solar tide between equinoxes. Conclusions are given in Section 4.

## 2  Results

### 2.1  Semidiurnal solar tide as measured by meteor radars

Specular meteor radars constitute an excellent tool to study winds in the mesosphere and lower thermosphere region (e.g., Hocking et al., 2001; McCormack et al., 2016). These remote sensing instruments continuously observe winds in the height

range extending from $\sim 75$ up to $105\ km$, with a typical vertical resolution of $2\ km$ (e.g., Stober et al., 2017). From these wind measurements, it is possible to extract detailed information about the mean winds and tides, as well as planetary waves and gravity waves (e.g., Hocking and Thayaparan, 1997; Hoffmann et al., 2007).

In this work, we have analyzed wind measurements provided by three meteor radars located at Andenes (69.3° N, 16° E), Juliusruh (54.6° N, 13.3° E) and Tavistock (42.3° N, 80.8° W). The tidal information has been estimated by means of a least

square technique. Assuming that the zonal ($u$) and meridional ($v$) winds result from the superposition of a mean wind ($U_0$ and $V_0$, respectively) plus oscillations of different periods, we independently fit the observations in each horizontal component with sinusoidal functions of periods $T_i$ equal to 24, 12 and 8 $h$ in order to account for the diurnal, semidiurnal and terdiurnal solar





Left: Andenes (69.3° N) / Middle: Juliusruh (54.6° N) / Right: Tavistock (42.3° N)

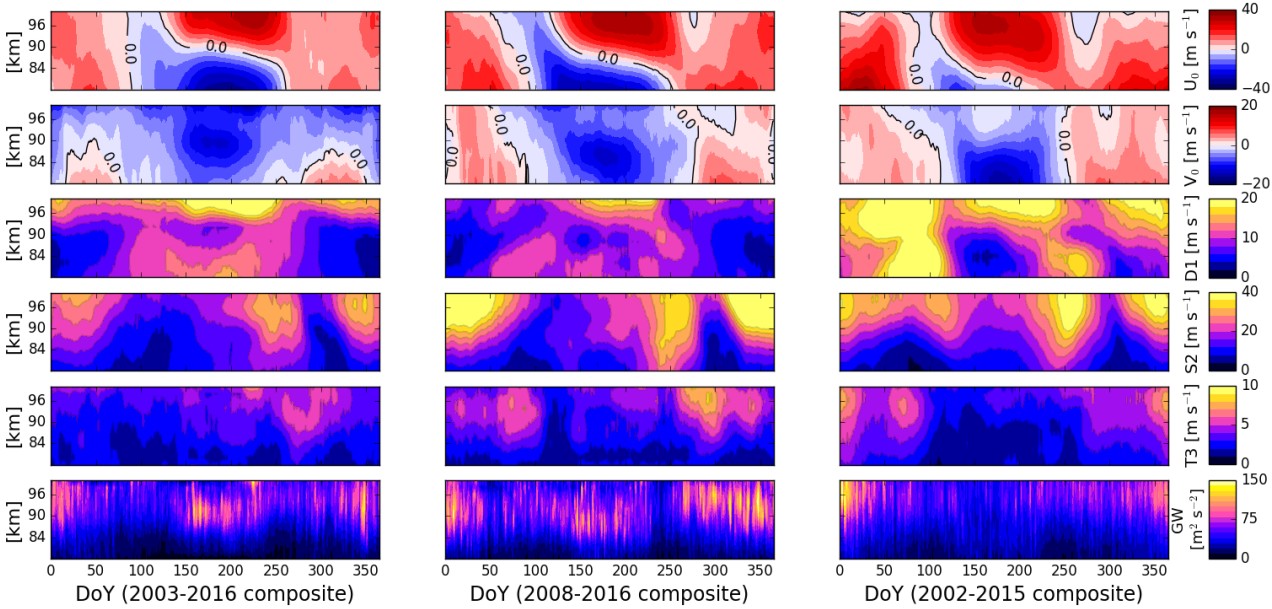

**Figure 1.** Composites of mean zonal ($U_0$) and meridional winds ($V_0$), diurnal, semidiurnal and terdiurnal solar tides (D1, S2 and T3, respectively), and GW kinetic energy over: (left) Andenes, (middle) Juliusruh and (right) Tavistock. The composites were determined for the entire yearly datasets available at the time this study was initiated.

tides,

$$(u,v) = (U_0, V_0) + \sum_{i=1}^{3} A_{(u,v)_i} \cos\left(2\pi \frac{(t - \phi_{(u,v)_i})}{T_i}\right); \tag{1}$$

where $A_{(u,v)_i}$ and $\phi_{(u,v)_i}$ represent the amplitude and phase, respectively, of the different tidal components. In this way, assuming that the tidal phases are stable within the selected running window, daily values of the mean winds, and the amplitude and phase of each tide were determined in bins of 21 days shifted by 1 day. Besides, information on the gravity wave activity is provided by the residuals of the fitting process, since they contain most of the wave-like perturbations different than tides and planetary waves.

It must be noted that from ground-based single station measurements it is not possible to decompose the observed tides into different wave numbers, which implies that we cannot differentiate between migrating and non-migrating tidal components.

In Fig. 1, we present a composite for the three sites considered in this study of the mean zonal ($U_0$) and meridional ($V_0$) winds, the total amplitude of the diurnal (D1), semidiurnal (S2), and terdiurnal (T3) solar tides, and a proxy for the gravity wave kinetic energy. The latter is estimated by adding and then dividing by 2 the squared residuals in the zonal and meridional components. The mean winds exhibit similar characteristics over the three sites: eastward zonal winds during winter, and a tilted wind reversal during the summer, with eastward winds above and westwards below and the height at which the wind





reversal is observed decreasing with latitude. The meridional winds blow toward the equator throughout the summer, and mainly poleward during the winter. The semidiurnal solar tide is the main interest of this study. Nevertheless, we are also presenting our results on the diurnal and terdiurnal solar tides to stress that the S2 tide dominates at middle and high latitudes, as previously reported (e.g., Manson et al., 1999; Hoffmann et al., 2010). By simple comparison, one can see that S2 shows

the largest amplitudes of the three tidal components. D1 does show significantly strong amplitudes during winter and early spring over Tavistock, but they are still a few $m\ s^{-1}$ smaller than those corresponding to S2. The seasonal behavior of S2 is characterized by strong amplitudes during the winter, that slowly ascending from lower to higher altitudes decrease during the spring time, to end with very low values during the summer. During the early fall, S2 amplitudes recover and reach values similar to (or even larger than) those observed during winter and that extend to lower altitudes than the rest of the year ($\sim 81$

$km$). Finally, during the fall the S2 tide abruptly decreases its amplitude at all height levels. This pronounced decrease is seen every year at the three locations, and extends for a period of $\sim$ 15 days or more, depending on the year.

The seasonal behavior of the gravity waves is different than that of the S2 tide. From the bottom panels of Fig. 1, it can be seen that GWs show significant activity during the winter at all three locations, but that it is considerable during summer only over Andenes and Juliusruh. On the other hand, GWs exhibit a clear pattern during equinox times over the three sites: there is

more activity during the fall than in the spring. This is clearest over Juliusruh, where an enhancement in the GW activity can be seen at approximately the same time the S2 tide abruptly decreases. The terdiurnal solar tide has a period of 8 $h$, which falls almost in the middle of the spectrum of typical GW periods. Since the wavelength information was not available for this study, we were not able to decompose the T3 tide into the 8 $h$ tidal wave and the GWs with that same period. Hence, to better assess the GW activity, the T3 tide must be included in the analysis. The seasonal variability of T3 presents some similarities with

that of the GWs, although the amplitude enhancement seen in the fall at around the same time S2 starts decreasing is more pronounced in the case of T3.

The time of occurrence of the S2 tide fall decrease varies from year to year. This can be deduced from Fig. 2, where we present the annual variability of the semidiurnal solar tide during the period 2008-2015, at all three locations. Over Andenes and Juliusruh, S2 starts decreasing earlier than on average during the years 2009, 2012 and 2013 (around day of the year [DoY]

270), and later during 2015 (around DoY 280). In the case of Tavistock, the years 2012 and 2013 show the earliest start of the S2 fall decrease ($\sim$ DoY 265), and again 2015 shows the latest ($\sim$ DoY 295). Besides an earlier start of the fall decrease, the year 2013 also shows the lowest S2 amplitudes previous to the decrease, over all three locations. Compared to autumn, there is more variability during the spring, specially at high latitudes, where two consecutive years may exhibit a difference of $\sim$ 20 days in the commencement of the reduction of S2 amplitudes. The duration of the fall decrease also varies from year to

year, and between different latitudes. At high latitudes, it tends to be $\sim$ 5-10 days longer than at middle latitudes (with 2009 showing the longest decrease), while over Julisuruh and Tavistock it extends for approximately the same amount of days (with the longest decrease observed during 2010 and 2011, respectively).





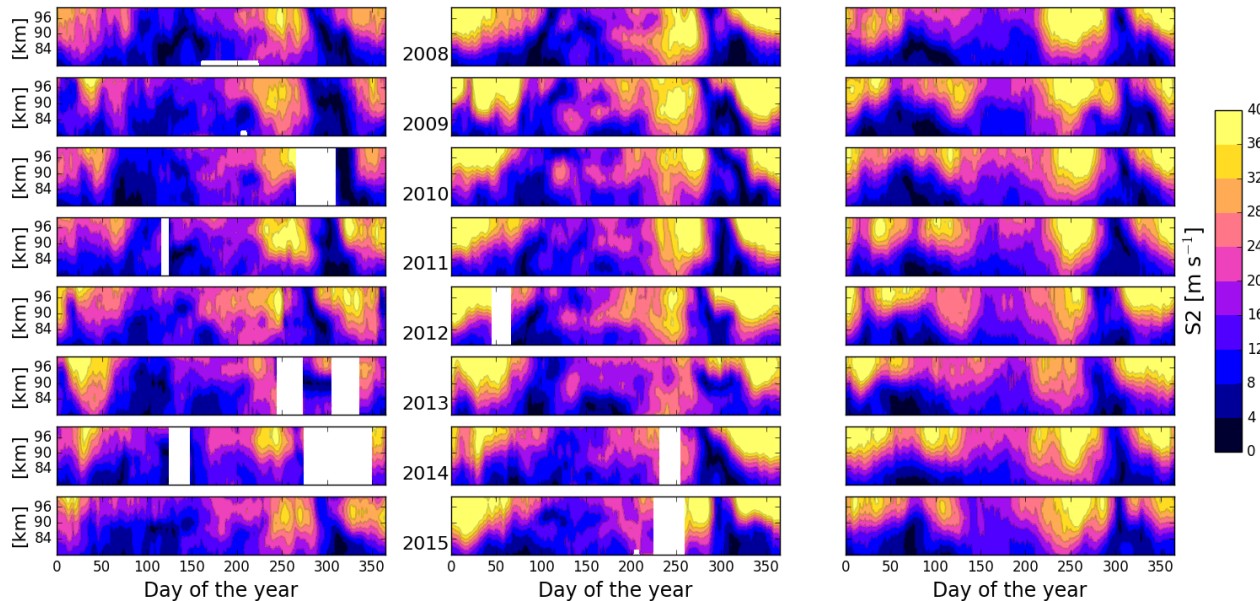

**Figure 2.** Semidiurnal solar tide (S2) observed during the time period (from top to bottom) 2008-2015, over: (left) Andenes, (middle) Juliusruh and (right) Tavistock. Data gaps are shown in white. The selected years correspond to the period with available data at the three locations.

## 2.2 Tides in the HAMMONIA model simulations

Motivated by our observations presented above, in this study we focus on the differences exhibited by the S2 tide between the spring and fall transition times. In order to investigate possible distinct behaviors of the different migrating and non-migrating semidiurnal tidal components, we have analyzed a 20-year simulation by the Hamburg Model of the Neutral and Ionized

5   Atmosphere (HAMMONIA).

HAMMONIA is a spectral model that consists of a vertical extension of the MAECHAM5 model (Giorgetta et al., 2006; Manzini et al., 2006) coupled to the MOZART3 chemical model (Kinnison et al., 2007). The atmospheric dynamics, chemistry and radiation processes are considered interactively from the Earth's surface up to $1.7 \times 10^{-7} \ hPa$ ($\sim 250 \ km$). Orographic gravity waves are parameterized according to Lott and Miller (1997), while non-orographic gravity waves are taken into

10  account following Hines (1997a, b). The 3-hour model outputs used in this study were obtained from a 20-year time-slice HAMMONIA simulation with constant boundary conditions typical for solar minimum and greenhouse gas concentrations of the 1990s, with a triangular truncation at wave number 31, corresponding to a resolution of $3.75°$ in latitude and longitude, and with 67 pressure levels. For a detailed description of the model, we refer the reader to Schmidt et al. (2006).




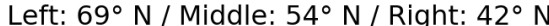

Left: 69° N / Middle: 54° N / Right: 42° N

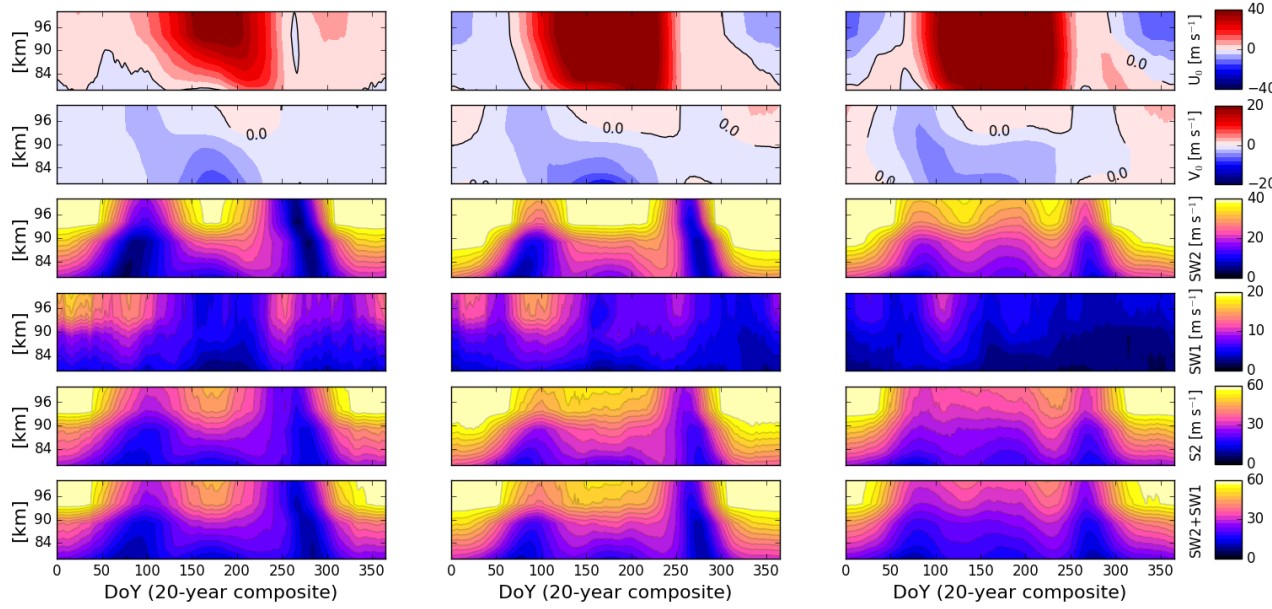

**Figure 3.** HAMMONIA composites of the simulated mean zonal ($U_0$) and meridional winds ($V_0$), migrating semidiurnal (SW2), non-migrating westward propagating wave number 1 semidiurnal (SW1), and total semidiurnal (S2) solar tides, and SW2+SW1, over: (left) Andenes, (middle) Juliusruh and (right) Tavistock.

To extract the tides and planetary waves, a similar fitting technique to that presented in the previous subsection was applied to the zonal and meridional winds simulated by HAMMONIA. The global coverage of the model outputs allows us to investigate different migrating and non-migrating components. Hence, at any given latitude and pressure level the following tides and planetary waves were fitted in a 30-day window shifted by 1 day,

$$\Omega = \Omega_0 + \sum_{l=1}^{4} \sum_{m=-3}^{3} B_{l,m} \cos\left(2\pi \frac{(t - m\lambda - \psi_{l,m})}{P_l}\right) +$$

$$+ \sum_{i=1}^{3} \sum_{s=-4}^{4} A_{i,s} \cos\left(2\pi \frac{(t - s\lambda - \phi_{i,s})}{T_i}\right). \tag{2}$$

Here, $\Omega$ represents either the zonal or the meridional wind, and $\Omega_0$ its mean; $s$, $m$ indicate the wave numbers (negative for a westward propagating wave) of tides and PWs, respectively; $\lambda$ is the geographic longitude; $A_{i,s}$ and $\phi_{i,s}$ are the respective amplitude and phase of tides with periods $T_i$ equal to 24, 12 and 8 hours; and $B_{l,m}$ and $\psi_{l,m}$ are the respective amplitude and phase of planetary waves with periods $P_l$ equal to 2, 5, 10 and 16 days.

Figure 3 shows composites for similar geographic coordinates than Andenes, Juliusruh and Tavistock, of the simulated mean zonal ($U_0$) and meridional ($V_0$) winds, total amplitude of the migrating semidiurnal (SW2), the non-migrating westward propagating wave number 1 semidiurnal (SW1), and the total semidiurnal (S2) solar tides, and $SW2 + SW1$. The altitude



range is the same as that of the meteor radar observations. However, in the case of the model the heights are approximated since its outputs are given in pressure levels. From Fig. 3, it can immediately be seen that the amplitudes of the mean zonal (meridional) wind are larger (smaller) than in the observations. Tidal amplitudes are also larger in the simulations than in the observations. The reversal of the mean zonal wind during summer is seen $\sim 10\ km$ lower than in the observations, a difference

that can be partially attributed to the approximated height calculated at each pressure level. Other discrepancies can be seen during winter at middle latitudes, where the simulated mean zonal winds are mainly westward. On the other hand, winter time mean zonal winds are eastward at high latitudes, in agreement with the observations. The simulated mean meridional winds are mainly equatorward during summer below $\sim 91\ km$, also in agreement with the observations. However, the poleward winds typically seen in meteor radar measurements during winter are not as evident in the simulations, specially at high latitudes.

The simulated SW2 tidal component exhibits similarities with the observed S2 tide: strong amplitudes in winter and both, the spring and fall transition times are characterized by a decrease in the activity of SW2, with the decrease seen in the fall being significantly more pronounced and occurring rapidly at all height levels. On the other hand, the amplitudes of SW2 during summer are considerably larger than those of the observed S2, while the SW2 fall decrease occurs earlier ($\sim 25$ days) and lasts a few more days than in the case of the observed S2.

The amplitude and seasonal behavior of SW1 are quite different. From Fig. 3, one can clearly see that the amplitudes of this tidal component never reach values larger than $20\ m\ s^{-1}$ and that the lowest values are mainly seen during the summer and the fall. In the spring, SW1 amplitudes over Andenes and Juliusruh remain approximately constant during the first half of the season, and slightly decrease during the second one. Over Tavistock, SW1 exhibits similar amplitude values throughout most of the spring. During the fall over Andenes and Juliusruh, SW1 amplitudes start decreasing around one week later than

SW2, and they reach the lowest values approximately one month after SW2 does. Over Tavistock, SW1 exhibits extremely low amplitudes the entire summer and fall. The rest of the semidiurnal tidal components extracted from HAMMONIA outputs have very small to negligible amplitudes, and are not shown here. The seasonal behavior of S2 and $SW2 + SW1$ are exactly the same (Fig. 3). The largest differences in amplitude are of only a few $m\ s^{-1}$, mainly during summer and early fall. Moreover, the amplitude and seasonal behavior of S2 are very similar to those of SW2, specially during the fall, which suggests that the

behavior of the total semidiurnal tide is mainly driven by SW2.

## 3 Discussion

The semidiurnal solar tide observed over Andenes, Juliusruh and Tavistock exhibits significant differences between the spring and fall transition times. These differences are reproduced by the HAMMONIA model. Although there are obvious discrepancies in the time of occurrence and duration of the fall decrease, as well as in the tidal amplitudes during the spring, the features

that characterize the tidal behavior during equinox times are similar in observations and simulations. Further, previous studies have already shown good agreement between observed tides and those simulated by HAMMONIA (e.g., Yuan et al., 2008). Therefore, we assume that one can use the analysis of the simulated tides to explain the behavior of S2 in the observations.



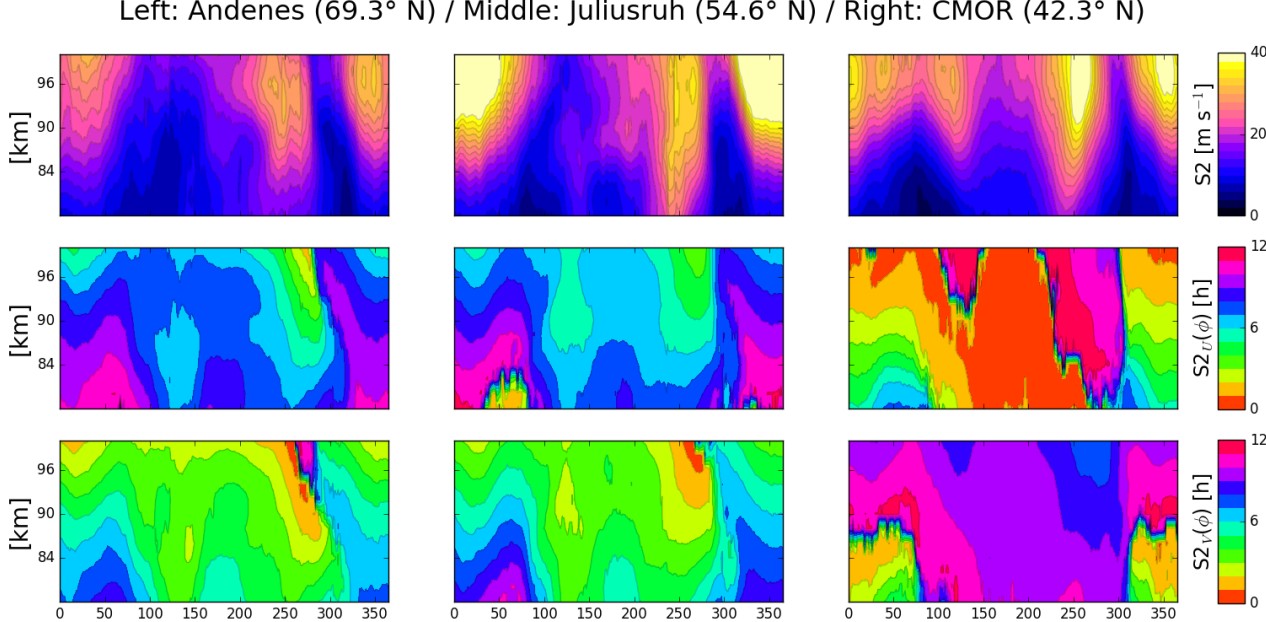

**Figure 4.** Composites of the semidiurnal solar tide (S2) and its phase in the zonal and meridional components, over: (left) Andenes, (middle) Juliusruh and (right) Tavistock.

Although both equinox time periods are characterized by a reduction in the activity of the S2 tide, the decrease observed in the fall is abrupt, more pronounced and it happens at all height levels at approximately the same time (Fig. 1). Furthermore, despite year-to-year changes in its duration and time of occurrence, the S2 fall decrease repeats every single year at both, middle and high latitudes (Fig. 2). Laskar et al. (2016) suggested that a possible explanation of this feature may be that

5   distinct migrating and non-migrating components of S2 behave differently during these periods. After extracting the tides from HAMMONIA simulations, we found that the simulated S2 tide is primarily determined by SW2 and SW1, which happen to present distinct behaviors during the spring and fall transition times. From Fig. 3, it can be seen that during the spring, SW2 decreases but SW1 maintains similar amplitude values or slightly decreases. During the fall, both tides decrease, although SW1 does it later. One can thus postulate that since SW1 does not decrease significantly during the spring, this allows the S2 tide

10  to maintain larger amplitudes than those observed during the fall. On the other hand, it seems that the influence of SW1 in the fall is not as relevant, given that this tidal component decreases later than SW2, but S2 decreases at the same time and with the same intensity as SW2. In other words, the observed S2 tide decreases during both equinox time periods because SW2 decreases, but during the spring the reduction of S2 is not as pronounced due to sustained higher amplitudes of SW1. In the fall, the observed decrease of S2 is more pronounced due to a more intense and longer decrease of SW2.

15      To further investigate the fall decrease, we have analyzed the phases of S2 and SW2. Forbes and Vial (1989) used model simulations to reproduce the phase change of the semidiurnal solar tide during equinox times, although they claimed that the




phase transition is more rapid during the spring. From Fig. 4, where we present a composite of the phases of the observed S2 in both horizontal components at the three locations considered in this work (we have included S2 total amplitudes for the purpose of a better comparison), one can see a clear shift in the phase of S2 at the time of the fall decrease. The late spring also shows changes in the phase, but these are not as pronounced and rapid as during the fall, specially over Andenes and Juliusruh,

where both, the zonal and meridional components show changes in the phase of $\sim 2\ h$ or more (over Tavistock, the phase changes in the meridional component are smaller). The behavior of the phase in the case of the simulated SW2 is similar: in both horizontal components, it exhibits a shift of $\sim 2\ h$ that manifests during the fall, when the amplitudes of SW2 abruptly decrease. This can be seen in Fig. 5, where besides the amplitude and phases of SW2, we also present the mean zonal wind (top panels) and the gravity waves (bottom panels) extracted from HAMMONIA, at Andenes, Juliusruh and Tavistock. The

latter were obtained in the same way as for the observations.

      Atmospheric tides are strongly influenced by mean winds and other types of waves (e.g., McLandress, 2002). Particularly, gravity waves propagating up from the troposphere cannot reach higher altitudes than those where their phase velocities match the wind velocity (e.g., Holton, 1983). However, during the fall transition time, the zonal wind in the stratosphere and mesosphere reverses from westward to eastward (see Fig. 5). Espy and Stegman (2002) showed that this reversal of the wind allows

GWs to freely propagate upwards and reach the MLT region. Therefore, it is possible that the change seen in the phase of S2 is the result of interactions between this tide and GWs (e.g., Fritts and Vincent, 1987). By comparing Fig. 1 and Fig. 4, one can see that the enhancement of the gravity wave activity detected during the fall occurs at approximately the same time the phase of S2 shifts significantly. In the case of HAMMONIA simulations, the gravity wave activity also enhances considerably during the fall (Fig. 5). Andenes does not show such a strong enhancement, although it is clear that there is more GW activity

during the fall than in the spring. GWs may naturally result from model simulations (e.g., Shepherd et al., 2000), but they strongly depend on the resolution of the model. As mentioned before, HAMMONIA runs at T31, which means that the smallest waves resolved by the model will have a length of more than $1000\ km$. This clearly imposes limitations to our analysis, since the gravity waves extracted from HAMMONIA simulations will only represent a small part of the actual GW spectrum. On the other hand, given that parameterized gravity waves can act only where they break (e.g., Sarin et al., 1996), a thoroughly

analysis of the mean flow is necessary in order to assess their impact on the tidal behavior. This is clearly out of the scope of the present study. Nonetheless, if one takes into consideration that the results based on the simulated gravity waves are only applicable to long scale waves, it follows that observations and model simulations reveal that the GW activity enhances at approximately the same time the semidiurnal solar tide exhibits a strong shift in its phase and a pronounced decrease in its amplitude. Consequently, one may speculate that the change seen during the fall in the phase of the S2 tide, as well as the

decrease of its amplitude, are partly due to interactions between SW2 and gravity waves. Future efforts will be focused on direct comparisons of the observed SW2 with GWs. For that, we plan to use multiple radars to separate SW2 and SW1 from the actual measurements, by means of a phase differencing technique (He et al., 2018).

      Given that the semidiurnal solar tide is produced mainly by solar heating of the ozone located in the stratosphere region (e.g., Chapman and Lindzen, 1970), the sudden and strong changes exhibited by S2 during the fall may also be related to changes

in the ozone concentration. Ozone variability has already been considered in previous studies in order to partially explain



the seasonal and longitudinal variability of S2 (e.g., Jacobi et al., 1999). Consequently, we also investigated the ozone levels simulated by the HAMMONIA model. We found no significant differences in the ozone concentration between the spring and fall transition times. Compared to the spring, the ozone levels are slightly reduced during the fall. However, this difference is not large enough so as to attribute the sudden decrease of S2 to a depletion of ozone.

**4  Conclusions**

Based on comparisons of meteor radar measurements with HAMMONIA model simulations, we showed that the differences exhibited by the semidiurnal solar tide (S2) observed at middle and high latitudes of the Northern Hemisphere between equinox times are mainly due to distinct behaviors of the migrating semidiurnal (SW2) and the non-migrating westward propagating wave number 1 semidiurnal (SW1) tidal components. Specifically, during the fall both, SW2 and SW1 decrease, while during

the spring time SW2 decreases but SW1 remains approximately constant or decreases only slightly. The decrease shown by SW1 during the fall occurs later than that of SW2 and S2, which indicates that the behavior of S2 is mainly driven by the migrating component. Nonetheless, the influence of SW1 is necessary to explain the behavior of S2 during the spring. Contributions by other semidiurnal tidal components were found to be very small to negligible.

In addition, we have shown that during the fall transition, at approximately the same time the observed S2 tide decreases, a

downward propagated phase shift (of $\sim 2\,h$ or more) can be seen in both horizontal components of this tide. The same feature was found in the model simulations, but in both horizontal components of SW2. Furthermore, our meteor radar observations show an increase in the gravity wave activity during this time period, possibly indicating that the phase shift, and the decrease in the amplitude of the semidiurnal solar tide, may be partly due to interactions between SW2 and gravity waves.

*Data availability.*  The Andenes and Juliusruh meteor radar data are available upon request from G. Stober. To get access to Tavistock meteor

radar data and/or HAMMONIA model simulations, please contact P. Brown and/or H. Schmidt.

*Competing interests.*  The authors declare that they have no competing interests.

*Acknowledgements.*  This work was supported by the Deutsche Forschungsgemeinschaft (DFG, German Research Foundation) under SPP
1788 (DynamicEarth) project CH 1482/1-1 (DYNAMITE).

The authors would like to thank Nick Pedatella from NCAR, and the members of the Tides matrix group at IAP for the fruitful discussions

25  that helped to improve the quality of this work.





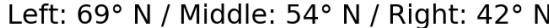
Left: 69° N / Middle: 54° N / Right: 42° N

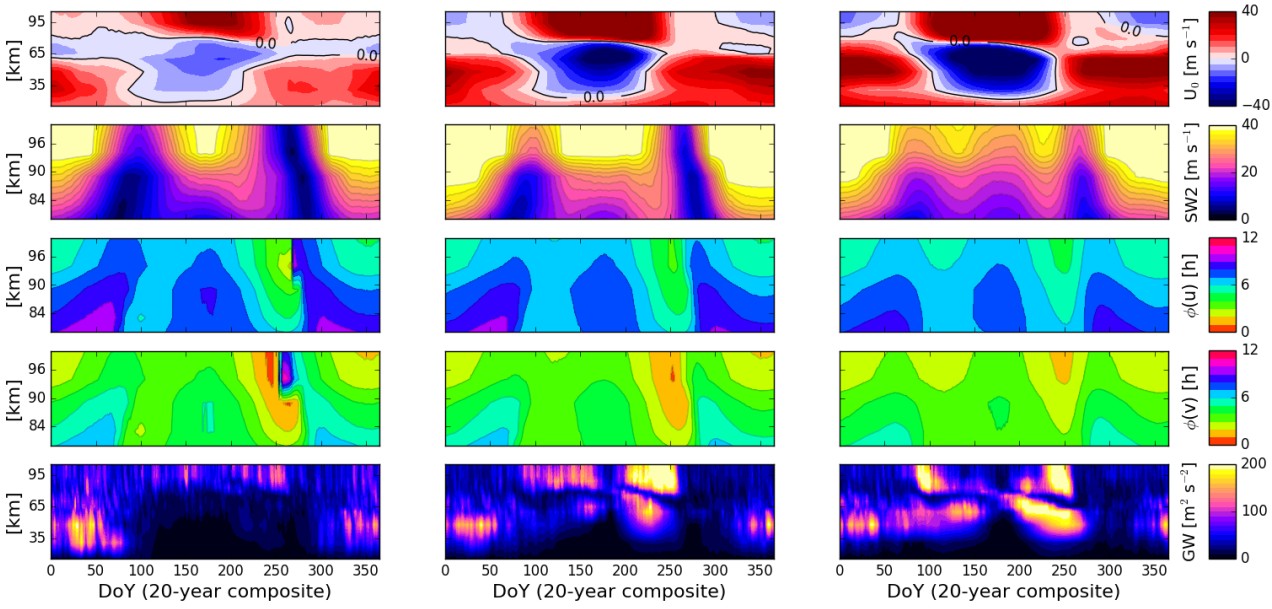

**Figure 5.** HAMMONIA composites of the simulated mean zonal ($U_0$) wind, the migrating semidiurnal solar (SW2) tide, SW2 phases in the zonal ($\phi(u)$) and meridional ($\phi(v)$) components, and gravity wave kinetic energy over: (left) Andenes, (middle) Juliusruh and (right) Tavistock. Notice the different height range in the case of $U_0$ and the GWs.

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
