# Peer review of "Semidiurnal solar tide differences between fall and spring transition times in the Northern Hemisphere"

_Annales Geophysicae, 2018_

## Referee Comment (RC1) · Anonymous Referee #1 · 4 May 2018

Review of the manuscript
 **"Semidiurnal solar tide differences between fall and spring transition times in the Northern Hemisphere"**
By J. Federico Conte et al. submitted to Ann. Geophys.
MS No.: angeo-2018-29

**General Comments:**

This paper is dedicated to research of semidiurnal solar tide (S2) behavior during the fall and spring transition times in Northern Hemisphere. Radar wind measurements by three meteor radars located at different mid-latitude sites have been used to investigate above-mentioned tide. It is observed an evident decrease of S2 during every autumn while the time of the decrease occurrence varies from year to year. There is the spring decrease as well but it is not so sudden. The next task that has been performed is to assess the contributions of different semidiurnal tidal components. To solve this problem the Hamburg Model of the Natural and Ionized Atmosphere (HAMMONIA) has been used. It is obtained that during the fall both migrating (SW2) and non-migrating westward propagating (SW1) semidiurnal tidal components decrease during the fall. During the spring, they behave in different ways. The observed behavior of the total semidiurnal tide S2 is mainly driven by superposition of SW2 and SW1 components. This is a good paper, which provides the readers with new information on the seasonal variations of semidiurnal tides. I believe that it will be accepted for publication in Ann. Geophys. after minor revision without an additional review.

I recommend that authors consider the following comments when revising the manuscript.

**Specific Comments:**

**Page 2, lines 1-2:** It should be noted that another possible source of non-migrating tides is a nonlinear interaction between migrating tides and stationary planetary waves (SPWs). The results obtained demonstrate that during seasonal transitions the SW2 and SW1 changes simultaneously. This fact indicates that they are not independent and connected through the nonlinear interaction with SPW1. It would be useful to include a short discussion of this possibility in the conclusion.

**Page 3, line 5 and page 6, line 4:** It is not clear why the different windows in analysis of measurements (21-days) and when the authors investigate the tides and planetary waves (30 days) have been used. The observations show a strong intra-seasonal variability of atmospheric tides and it is not correctly to use different bins in the analysis. For a future, I would like to suggest the complex Morlet wavelet transform to investigate the intra-seasonal variability of atmospheric tides.

**Page 6, line 12:** It would be useful to explain shortly the difference between the total semidiurnal tide (S2) and SW2+SW1 (at least when we consider the results of simulation).

**Technical corrections:**

**Page 2, line9 and page 10, line 17:** It is better to use "time interval" instead of "time period".

---

## Referee Comment (RC2) · Anonymous Referee #2 · 8 May 2018

**Reviewer comments on the submitted manuscript: "Semidiurnal solar tide differences between fall and spring transition times in the Northern Hemisphere"**

The current paper focuses on differences in variability of semidiurnal solar tide (S2) between autumn and spring in the Northern Hemisphere. The differences were first described using wind data observed by meteor radars at three stations: one at high latitude (Andenes) and two at mid-latitudes (Juliusruh and Tavistock). In brief, S2 was found to decrease suddenly at all observed altitudes in autumn, while in spring S2 decreses more gradually and the decrease occurs earlier at lower altitudes than at higher altitudes. In order to explain these differences, the authors considered contributions from dominant semidiurnal tidal components (SW1 and SW2) provided by HAMMONIA simulation. The authors found that differences in variabilities of both SW1 and SW2 mostly lead to different variabilities of S2 during autumn and spring. In addition, gravity wave (GW) activity observed by meteor radars is stronger in autumn than in spring. This, as suggested by the authors, may also contribute to differences in S2 behavior via GW-tide interaction.

The paper is scientifically interesting. It is generally well written and clearly structured. It also has an adequate length and pertinent title and abstract.

**General comments:**

1. The introduction mentions only one possible reason, which could lead to S2 differences between autumn and spring (tide-tide interaction). Another possible reason, as you dicussed later in your manuscript, could be GW-tide interaction. Although it is not the main topic of the current paper, I think GW-tide interaction should be briefly mentioned in the introduction.

2. To estimate the tidal information from meteor radar measurements, a running window of 21 days was used. For HAMMONIA simulation, a 30-day window was used. Can you please explain: (1) the reason why 21 days and 30 days were chosen? and (2) why is the window for wind observations different from the window for simulated wind?

Further, extracting tides from HAMMONIA simulation took into account PWs, but extracting tides from radar measurements did not consider PWs. This difference should be explained in the manuscript.

3. For all figures in the manuscript, please add a vertical grid or at least 2 vertical lines for each spring and autumn. This will help the readers very much to follow the variability (decrease) of tidal components that you described in the text.

4. The fall decrease occurs earlier in HAMMONIA simulation than in the observations. This can be seen for all 3 locations and very clearly for Juliusruh and Tavistock. Please describe and explain this fact in your paper.

**Specific comments:**

Below, the first number is the page number and the second number is the line number or line numbers, separated by a forward slash. For example, 2/5 refers to page 2, line 5; 3/10-12 means page 3 lines 10 to 12.

1. 1/19: PW → PWs

2. 1/20: GW → GWs

3. 1/22: "they have typical periods ...". Some rewording may be needed. My suggestion: "The most dominant tide components have periods .."

4. 2/14: It is helpful for the readers to introduce again the abbreviations (SW2 and SW1) here in parentheses

5. 3/13: "The mean winds .." → "The zonal mean winds .."

6. 4/14: Which part of GW spectrum can be seen by your measurements? Please specify the observed GW spectrum.

7. 4/23-26: Do you have any explanation for the ealier decrease during the years 2009, 2012, 2013 and the lower amplitude during 2013?

8. 4/28: What is the reason of more variability during the spring?

9. 4/30: Can you please explain why the duration is longer at high latitudes than at middle latitudes?

10. 5/10: Is it possible to turn off the GW parameterizations and see how much GW-tide interaction influences S2 variability?

11. 7/14: Do you also see the annual variability in HAMMONIA simulation?

12. 7/14: Please mention that the fall decrease in simulation occurs ealier than in observations. Further, the simulated S2 amplitude is higher than observed S2 amplitude. Can you please also comment on that?

13. 7/28: "These differences are reproduced .. model" → "These differences are reproduced .. model to a certain extent"

14. 8/0: Title of Fig. 4: CMOR → Tavistock?

15. 9/8: For observations, you showed the phase analysis for S2 (Fig. 4). For simulation, why don't you show the phase analysis for the same S2? why did you choose SW2 instead?

16. 9/25: I agree with the authors that GW-tide interaction requires a thorough analysis, given that not only the simulation, but also your observations contain only a certain part of the GW spectrum, and different parts of the GW spectrum can interact differently with tides. The interaction of other parts of the GW spectrum with S2 cannot be estimated here and may also influence the S2 variability. Maybe you should add one sentence here to clarify that fact.

17. 9/27: "the GW activity" $\rightarrow$ "the long-wave GW activity"

18. 9/30: Would you conclude that long-wave GWs suppress the migrating semidiurnal tide SW2 amplitude?

---

## Author Comment (AC2) · 1 Jun 2018

**Response to anonymous referee #2**

**"Semidiurnal solar tide differences between fall and spring transition times in the Northern Hemisphere2**

By J. Federico Conte et al. submitted to Ann. Geophys.

MS: angeo-2018-29

The current paper focuses on differences in variability of semidiurnal solar tide (S2) between autumn and spring in the Northern Hemisphere. The differences were first described using wind data observed by meteor radars at three stations: one at high latitude (Andenes) and two at mid-latitudes (Juliusruh and Tavistock). In brief, S2 was found to decrease suddenly at all observed altitudes in autumn, while in spring S2 decreases more gradually and the decrease occurs earlier at lower altitudes than at higher altitudes. In order to explain these differences, the authors considered contributions from dominant semidiurnal tidal components (SW1 and SW2) provided by HAMMONIA simulation. The authors found that differences in variabilities of both SW1 and SW2 mostly lead to different variabilities of S2 during autumn and spring. In addition, gravity wave (GW) activity observed by meteor radars is stronger in autumn than in spring. This, as suggested by the authors, may also contribute to differences in S2 behavior via GW-tide interaction.

The paper is scientifically interesting. It is generally well written and clearly structured. It also has an adequate length and pertinent title and abstract.

**We would like to thank this anonymous referee for taking the time to read and review our paper. The response to each comment can be found below. Also, we have attached a new version of the manuscript so the referee can see the applied changes.**

*General comments:*
1. The introduction mentions only one possible reason, which could lead to S2 differences between autumn and spring (tide-tide interaction). Another possible reason, as you discussed later in your manuscript, could be GW-tide interaction. Although it is not the main topic of the current paper, I think GW-tide interaction should be briefly mentioned in the introduction.
**R: thank you for this comment. We have added a sentence with a proper reference on this matter.**

2. To estimate the tidal information from meteor radar measurements, a running window of 21 days was used. For HAMMONIA simulation, a 30-day window was used. Can you please explain: (1) the reason why 21 days and 30 days were chosen? and (2) why is the window for wind observations different from the window for simulated wind?
**R: thank you for this comment. Now, we explain in the manuscript the reason for selecting 21-day and 30-day windows. Briefly, the number of daily unknowns to be determined in HAMMONIA simulations, at each pressure level, is 111. Squared, this is 12,321. A fitting window of 21 days would be large enough (and would coincide with the window selected for the observations) to obtain a solution after applying least squares (21 [days] x 8 [time points] x 96 [longitude points] = 16,128). However, in order to reduce the error of the fitting process and avoid some numerical artifacts, we had to use a window of 30 days. On the other hand, using a 30-day window for the observations would significantly smooth the tidal behavior (from previous studies of the authors of this**

**manuscript and other researchers, we know that a 21-day window is still good enough for climatological studies such as the one presented here).**

Further, extracting tides from HAMMONIA simulation took into account PWs, but extracting tides from radar measurements did not consider PWs. This difference should be explained in the manuscript. **R: thank you very much for this comment. We also fitted HAMMONIA wind simulations without explicitly considering the PWs (as in the case of meteor radar measurements). The mean winds and tides determined in this second case were very similar to those obtained with the explicit inclusion of PWs, and hence we decided not to consider the results of this second fitting. We now explain this in the manuscript.**

3. For all figures in the manuscript, please add a vertical grid or at least 2 vertical lines for each spring and autumn. This will help the readers very much to follow the variability (decrease) of tidal components that you described in the text. **R: thank you for this comment. We have added vertical white dashed lines in all our figures for days of the year 90 and 275, so the readers can better understand our results.**

4. The fall decrease occurs earlier in HAMMONIA simulation than in the observations. This can be seen for all 3 locations and very clearly for Juliusruh and Tavistock. Please describe and explain this fact in your paper. **R: thanks. This is mentioned in the manuscript. Nevertheless, we have modified a few sentences and added new ones so this is clearer for the readers.**

*Specific comments:*
Below, the first number is the page number and the second number is the line number or line numbers, separated by a forward slash. For example, 2/5 refers to page 2, line 5; 3/10-12 means page 3 lines 10 to 12.

1. 1/19: PW → PWs
**R: thanks. We have made this correction.**

2. 1/20: GW → GWs
**R: thanks. We have corrected this.**

3. 1/22: "they have typical periods ...". Some rewording may be needed. My suggestion: "The most dominant tide components have periods .."
**R: thanks. We have re-written this sentence in a different way.**

4. 2/14: It is helpful for the readers to introduce again the abbreviations (SW2 and SW1) here in parentheses
**R: thanks. We have included the acronyms again.**

5. 3/13: "The mean winds .." → "The zonal mean winds .."
**R: thanks for this. We have corrected the sentence.**

6. 4/14: Which part of GW spectrum can be seen by your measurements? Please specify the observed GW spectrum.
**R: thanks. Now we have specified that the meteor radar observations allow us to see GWs with periods larger than 2 hours.**

7. 4/23-26: Do you have any explanation for the earlier decrease during the years 2009, 2012, 2013 and the lower amplitude during 2013?

**R: sadly, no. We have investigated some possible agents that might be causing these differences (e.g., solar activity levels, proximity to strong warmings the following winter, etc.) but found nothing convincing. We plan to further investigate the year-to-year variability of S2.**

8. 4/28: What is the reason of more variability during the spring?
**R: again, sadly we do not have a convincing answer to this question. We believe that the observed variability during the spring could be connected, e.g., to late warmings. Late stratospheric warmings have been observed during March/April of, e.g., 2005 and 2015. We think this late warmings might be partially introducing more variability in the MLT region. But, again we need to further investigate this.**

9. 4/30: Can you please explain why the duration is longer at high latitudes than at middle latitudes?
**R: as in the case of point 7., we still cannot explain this. That is one of the reasons why we want to estimate the impact of SW1 directly from the observations (mentioned in the manuscript) in order to see if this longer duration at high latitudes has some connection with the planetary wave activity.**

10. 5/10: Is it possible to turn off the GW parameterizations and see how much GW-tide interaction influences S2 variability?
**R: in theory, it could be done. However, we have used HAMMONIA simulations that were already stored in Hamburg MPI servers. To run new simulations would be time consuming and sadly, H. Schmidt is extremely busy with other projects right now. Nonetheless, in future studies we plan to "play" with different GW parameterizations using other models (such as WACCM-X and CMAT-2).**

11. 7/14: Do you also see the annual variability in HAMMONIA simulation?
**R: No. We see very similar behaviors during all the 20 years used in the study.**

12. 7/14: Please mention that the fall decrease in simulation occurs earlier than in observations. Further, the simulated S2 amplitude is higher than observed S2 amplitude. Can you please also comment on that?
**R: These features are already mentioned in the manuscript. However, we have added new mentions of these facts in other parts of the manuscript in order to make it more clear.**

13. 7/28: "These differences are reproduced .. model" → "These differences are reproduced .. model to a certain extent"
**R: thanks. We have applied the suggested change.**

14. 8/0: Title of Fig. 4: CMOR → Tavistock?
**R: thanks. We have changed the title accordingly.**

15. 9/8: For observations, you showed the phase analysis for S2 (Fig. 4). For simulation, why don't you show the phase analysis for the same S2? why did you choose SW2 instead?
**R: well, for two main reasons. Firstly, because we fitted each tidal component separately. Hence, mixing the phases of all tidal components contributing to S2 would be misleading. Secondly, because after realizing that SW2 strongly dominates the behavior of S2 during the fall, we mainly focused our study on this component (in the simulations).**

16. 9/25: I agree with the authors that GW-tide interaction requires a thorough analysis, given that not only the simulation, but also your observations contain only a certain part of the GW spectrum, and different parts of the GW spectrum can interact differently with tides. The interaction of other parts of the GW spectrum with S2 cannot be estimated here and may also influence the S2 variability. Maybe you should add one sentence here to clarify that fact.

**R: thanks. We have added a sentence on this matter.**

17. 9/27: "the GW activity" → "the long-wave GW activity"
**R: thanks for this comment. We believe that in the context of the sentence it is clear we are talking about long-wave GWs. Hence, we did not apply this change.**

18. 9/30: Would you conclude that long-wave GWs suppress the migrating semidiurnal tide SW2 amplitude?
**R: we think that the GW-tide interaction could be one of the causes of the fall decrease of SW2. However, we don't have sufficient evidence yet to firmly conclude this, and that is why we are only suggesting that GWs could be involved.**

---

## Author Response (AR1)

**Response to Topical Editor**

**"Semidiurnal solar tide differences between fall and spring transition times in the Northern Hemisphere2**

By J. Federico Conte et al. submitted to Ann. Geophys.

MS: angeo-2018-29

Comments to the Author:

The editor requests the authors to add page and line numbers to indicate the newly added or modified text in the revised manuscript.
**R: Thanks for this comment. We apologize for our mistake. We should have indicated where the changes were applied. We now indicate the page and line of the modifications introduced in the revised manuscript. Please see new replies to the reviewers.**

Also, the use of two different window lengths (see Rev#1, second comment and Rev#2 General comment 2) need to be discussed by the authors in more detail, e.g., geophysical pros/cons in addition to technical pros/cons.
**R: Thank you. We have checked our code for processing the HAMMONIA simulations and found a small bug that was responsible for the artifacts we were seeing with windows shorter than 30 days. We used the corrected code and reprocessed the HAMMONIA outputs using a 21-day window. After analyzing the new results, we found very small differences with respect to the results using a 30-day window (that can be seen in the new figures of the manuscript, if compared with the original ones), namely, a decrease in the amplitudes of the GWs extracted from HAMMONIA (of around 10%), and a shift of 1-2 days in the timing of the fall and spring transitions. These differences clearly do not modify the conclusions of our study. Nevertheless, we now specify that the averaging window is the same for both, observations and simulations (21 days), and give the reasons for selecting this length. Please see new replies to the reviewers.**

Non-public comments to the author:

The authors have responded to each of the reviewers comments, at few places the responses cannot fully clarify the concerns.
**R: we have re-written some of the responses to the reviewers with answers we think better address their concerns. Please see new replies to the reviewers.**

At many places the authors did not indicate where in the newly uploaded and revised manuscript (page and line numbers) their answers have been placed. It is urgently recommended doing so, otherwise it is not possible to judge if the answers are adequate.
These are the case for:
Reviewer 1: for all three comments.

Reviewer 2: "General comments" 1, 2, and 4 and "Specific comments" 6, 12, 16 .

**R: Thanks. We now indicate page number and lines of the new or corrected sentences. Please see new replies to the reviewers.**

Rev#2, Specific comment 17: Please apply long-wave if it is not wrong. Apparently, that "long-wave" was meant has not been clear to all readers.

**R: we have added the term "long-wave". Please see in the revised manuscript: page 10, line 5.**

Rev#1, second comment AND Rev#2 General comment 2: The use of different window lengths has been discussed and questioned by both reviewers. The authors explain the longer window for the model results "to reduce numerical artefacts". However, the concern was about a physical difference that is introduced when using 2 different window lengths. How the authors deal this aspect is not discussed. Possible cons using the two windows needs to be mentioned and discussed in the paper, as well as arguments need to be addressed in case the authors still believe that the two window sizes are the best choices.

**R: thank you for this comment. As we mentioned above, we have corrected our code and reprocessed the HAMMONIA simulations using the same window length as with the observations (i.e., 21 days). With the exception of a change of ~10% in the amplitudes of the long-scale GWs extracted from HAMMONIA and a shift of 1-2 days in the time of occurrence of the fall and spring transitions, we haven't found differences with respect to the case of the 30-day averaging window. Nevertheless, as the editor correctly pointed out, using different size windows could introduce different physics in the comparisons. Hence, we now base our analysis on the 21-day window for both, observations and simulations.**
**The reason for selecting a 21-day window is explained in the manuscript; briefly, given the number of unknowns to be determined from the simulations we need to consider a 21-day window in order to obtain a robust solution after applying least squares (please see in the revised manuscript: page 6, lines 9-11).**

**Response to anonymous referee #1**

**"Semidiurnal solar tide differences between fall and spring transition times in the Northern Hemisphere2**

By J. Federico Conte et al. submitted to Ann. Geophys.

MS: angeo-2018-29

*General Comments:*
This paper is dedicated to research of semidiurnal solar tide (S2) behavior during the fall and spring transition times in Northern Hemisphere. Radar wind measurements by three meteor radars located at different mid-latitude sites have been used to investigate above-mentioned tide. It is observed an evident decrease of S2 during every autumn while the time of the decrease occurrence varies from year to year. There is the spring decrease as well but it is not so sudden. The next task that has been performed is to assess the contributions of different semidiurnal tidal components. To solve this problem the Hamburg Model of the Natural and Ionized Atmosphere (HAMMONIA) has been used. It is obtained that during the fall both migrating (SW2) and non-migrating westward propagating (SW1) semidiurnal tidal components decrease during the fall. During the spring, they behave in different ways. The observed behavior of the total semidiurnal tide S2 is mainly driven by superposition of SW2 and SW1 components. This is a good paper, which provides the readers with new information on the seasonal variations of semidiurnal tides. I believe that it will be accepted for publication in Ann. Geophys. after minor revision without an additional review.
I recommend that authors consider the following comments when revising the manuscript.

**We would like to thank this anonymous referee for taking the time to read and revise our manuscript. Below, you can find the response to each comment.**

*Specific Comments:*

**Page 2, lines 1-2:** It should be noted that another possible source of non-migrating tides is a nonlinear interaction between migrating tides and stationary planetary waves (SPWs). The results obtained demonstrate that during seasonal transitions the SW2 and SW1 changes simultaneously. This fact indicates that they are not independent and connected through the nonlinear interaction with SPW1. It would be useful to include a short discussion of this possibility in the conclusion.
**R: thank you for this comment. We have added a paragraph explaining why we did not consider non-linear interaction between tides and stationary planetary waves in our analysis (no substantial planetary wave activity was found in the model during the fall and spring times). Please see in the revised manuscript: page 7, lines 33-34, and page 8, lines 1-2.**
**Nevertheless, we would like to point out that some of the co-authors have recently published an article where they study non-linear interactions between the semidiurnal solar tide and SPW1 using wavelet transforms, but during sudden stratospheric warming events (He et al., 2017 https://agupubs.onlinelibrary.wiley.com/doi/abs/10.1002/2017JA024630).**

**Page 3, line 5 and page 6, line 4:** It is not clear why the different windows in analysis of measurements (21-days) and when the authors investigate the tides and planetary waves (30 days) have been used. The observations show a strong intra-seasonal variability of atmospheric tides and it is not correctly to use

different bins in the analysis. For a future, I would like to suggest the complex Morlet wavelet transform to investigate the intra-seasonal variability of atmospheric tides.

**R: Thank you very much for this comment. We explain now in the manuscript the reason for selecting a 21-day window: the number of daily unknowns to be determined from the model outputs, at each pressure level, is 111. Squared, this is 12,321. A fitting window of 21 days is large enough to obtain a robust solution after applying least squares (21 [days] x 8 [time points] x 96 [longitude points] = 16,128). Please see in the revised manuscript: page 6, lines 9-11.**

**Furthermore, we have reprocessed HAMMONIA outputs using a 21-day window and present now our analysis based on this averaging window length. Both, simulations and observations are now analyzed using the same averaging window size: 21 days. Before, we hadn't done that because we had found that some artifacts were produced (in the simulations outputs) when using a window smaller than 30 days. We revised our code and found a bug responsible for this. Hence, we re-analyzed HAMMONIA outputs using a 21-day window and found no differences (compared to the results obtained with a 30-day window), with the exception of ~10% smaller amplitudes in the GWs extracted from the simulations and a shift of 1-2 days in the time of occurrence of the S2 fall and spring transitions. In other words, our conclusions remain the same, but now are based on the correct comparison. Please see in the revised manuscript: new Figures 3, 4 and 5.**

**Page 6, line 12:** It would be useful to explain shortly the difference between the total semidiurnal tide (S2) and SW2+SW1 (at least when we consider the results of simulation).

**R: Thanks for this comment. We have added a couple of sentences describing the difference between the total S2 tide and SW2+SW1. Please see in the revised manuscript: page 7, lines 5-6.**

*Technical corrections:*

**Page 2, line9 and page 10, line 17:** It is better to use "time interval" instead of "time period".

**R: Thanks for this suggestion. We have changed "time period" to "time interval". Please see in the revised manuscript: page 2, lines 6 and 9.**

**Response to anonymous referee #2**

**"Semidiurnal solar tide differences between fall and spring transition times in the Northern Hemisphere2**

By J. Federico Conte et al. submitted to Ann. Geophys.

MS: angeo-2018-29

The current paper focuses on differences in variability of semidiurnal solar tide (S2) between autumn and spring in the Northern Hemisphere. The differences were first described using wind data observed by meteor radars at three stations: one at high latitude (Andenes) and two at mid-latitudes (Juliusruh and Tavistock). In brief, S2 was found to decrease suddenly at all observed altitudes in autumn, while in spring S2 decreases more gradually and the decrease occurs earlier at lower altitudes than at higher altitudes. In order to explain these differences, the authors considered contributions from dominant semidiurnal tidal components (SW1 and SW2) provided by HAMMONIA simulation. The authors found that differences in variabilities of both SW1 and SW2 mostly lead to different variabilities of S2 during autumn and spring. In addition, gravity wave (GW) activity observed by meteor radars is stronger in autumn than in spring. This, as suggested by the authors, may also contribute to differences in S2 behavior via GW-tide interaction.

The paper is scientifically interesting. It is generally well written and clearly structured. It also has an adequate length and pertinent title and abstract.

**We would like to thank this anonymous referee for taking the time to read and review our paper. The response to each comment can be found below. Also, we have attached a new version of the manuscript so the referee can see the applied changes.**

*General comments:*
1. The introduction mentions only one possible reason, which could lead to S2 differences between autumn and spring (tide-tide interaction). Another possible reason, as you discussed later in your manuscript, could be GW-tide interaction. Although it is not the main topic of the current paper, I think GW-tide interaction should be briefly mentioned in the introduction.
**R: thank you for this comment. We have added a sentence with a proper reference on this matter. Please see in the revised manuscript: page 2, lines 15-17.**

2. To estimate the tidal information from meteor radar measurements, a running window of 21 days was used. For HAMMONIA simulation, a 30-day window was used. Can you please explain: (1) the reason why 21 days and 30 days were chosen? and (2) why is the window for wind observations different from the window for simulated wind?
**R: Thank you very much for this comment. Firstly, we now explain in the manuscript the reason for selecting a 21-day window: the number of daily unknowns to be determined from the model outputs, at each pressure level, is 111. Squared, this is 12,321. A fitting window of 21 days is large enough to obtain a robust solution after applying least squares (21 [days] x 8 [time points] x 96 [longitude points] = 16,128). Please see in the revised manuscript: page 6, lines 9-11.**

**Secondly, we have reprocessed HAMMONIA outputs using a 21-day window and present now our analysis based on this averaging window size. Both, simulations and observations are now analyzed using the same averaging window size: 21 days. Before, we hadn't done that because we had found that some artifacts were produced (in the simulations outputs) when using a window smaller than 30 days. We revised our code and found the bug responsible for this. Hence, we re-analyzed HAMMONIA outputs using a 21-day window and found no differences (compared to the results obtained with a 30-day window), with the exception of ~10% smaller amplitudes in the GWs extracted from the simulations and a shift of 1-2 days in the time of occurrence of the fall and spring transitions. In other words, our conclusions remain the same, but now are based on the correct comparison, given that we are using the same averaging window size for both, observations and simulations. Please see in the revised manuscript: new Figures 3, 4 and 5.**

Further, extracting tides from HAMMONIA simulation took into account PWs, but extracting tides from radar measurements did not consider PWs. This difference should be explained in the manuscript. **R: thank you very much for this comment. We also fitted HAMMONIA wind simulations without explicitly considering the PWs (as in the case of meteor radar measurements). The mean winds and tides determined in this second case were very similar to those obtained with the explicit inclusion of PWs, and hence we decided not to consider the results of this second fitting. We now explain this in the manuscript. Please see in the revised manuscript: page 6, line 11, and page 7, lines 1-2.**

3. For all figures in the manuscript, please add a vertical grid or at least 2 vertical lines for each spring and autumn. This will help the readers very much to follow the variability (decrease) of tidal components that you described in the text. **R: thank you for this comment. We have added vertical white dashed lines in all our figures for days of the year 90 and 275, so the readers can better understand our results. Please see in the revised manuscript the new figures and also the captions, where we indicate that the vertical white dashed lines mark the days of the year 90 and 275.**

4. The fall decrease occurs earlier in HAMMONIA simulation than in the observations. This can be seen for all 3 locations and very clearly for Juliusruh and Tavistock. Please describe and explain this fact in your paper. **R: thanks. This is mentioned in the manuscript. Nevertheless, we have modified a few sentences and added new ones so this is clearer for the readers. Please see in the revised manuscript: page 7, lines 18-21, and page 8, lines 6-7.**

*Specific comments:*
Below, the first number is the page number and the second number is the line number or line numbers, separated by a forward slash. For example, 2/5 refers to page 2, line 5; 3/10-12 means page 3 lines 10 to 12.

1. 1/19: PW → PWs
**R: thanks. We have made this correction. Please see in the revised manuscript: page 1, line 19.**

2. 1/20: GW → GWs
**R: thanks. We have corrected this. Please see in the revised manuscript: page 1, line 20.**

3. 1/22: "they have typical periods ...". Some rewording may be needed. My suggestion: "The most dominant tide components have periods .."
**R: thanks. We have re-written this sentence in a different way. Please see in the revised manuscript: page 1, lines 22-23.**

4. 2/14: It is helpful for the readers to introduce again the abbreviations (SW2 and SW1) here in parentheses

**R: thanks. We have included the acronyms again. Please see in the revised manuscript: page 2, lines 14-15.**

5. 3/13: "The mean winds .." → "The zonal mean winds .."

**R: thanks for this. We have corrected the sentence. Please see in the revised manuscript: page 3, line 13.**

6. 4/14: Which part of GW spectrum can be seen by your measurements? Please specify the observed GW spectrum.

**R: thanks. Now we have specified that the meteor radar observations allow us to see GWs with periods larger than 2 hours. Please see in the revised manuscript: page 4, lines 17-18.**

7. 4/23-26: Do you have any explanation for the earlier decrease during the years 2009, 2012, 2013 and the lower amplitude during 2013?

**R: sadly, no. We have investigated some possible agents that might be causing these differences (e.g., solar activity levels, proximity to strong warmings the following winter, etc.) but found nothing convincing. We are currently investigating the influence of polar jet oscillations on thermal tides and we think it might be related to that, but unfortunately we cannot prove this yet.**

8. 4/28: What is the reason of more variability during the spring?

**R: again, sadly we do not have an answer to this question. We believe that the observed variability during the spring could be connected, e.g., to late warmings. Late stratospheric warmings have been observed during March/April of, e.g., 2005 and 2015. We think this late warmings might be partially introducing more variability in the MLT region. But, again we need to further investigate this.**

9. 4/30: Can you please explain why the duration is longer at high latitudes than at middle latitudes?

**R: as in the case of point 7., we still cannot explain this. That is one of the reasons why we also want to estimate the impact of SW1 directly from the observations (mentioned in the manuscript) in order to see if this longer duration at high latitudes has some connection with the planetary wave activity.**

10. 5/10: Is it possible to turn off the GW parameterizations and see how much GW-tide interaction influences S2 variability?

**R: in theory, it could be done. However, we have used HAMMONIA simulations that were already stored in Hamburg MPI servers. To run new simulations would be time consuming and sadly, H. Schmidt is extremely busy with other projects right now. Nonetheless, in future studies we plan to "play" with different GW parameterizations using other models (such as WACCM-X and CMAT-2).**

11. 7/14: Do you also see the annual variability in HAMMONIA simulation?

**R: No. We see very similar behaviors during all the 20 years used in the study.**

12. 7/14: Please mention that the fall decrease in simulation occurs earlier than in observations. Further, the simulated S2 amplitude is higher than observed S2 amplitude. Can you please also comment on that?

**R: These features are already mentioned in the manuscript.**

13. 7/28: "These differences are reproduced .. model" → "These differences are reproduced .. model to a certain extent"
**R: thanks. We have applied the suggested change. Please see in the revised manuscript: page 8, line 5.**

14. 8/0: Title of Fig. 4: CMOR → Tavistock?
**R: thanks. We have changed the title accordingly. Please see in the revised manuscript: Fig. 4.**

15. 9/8: For observations, you showed the phase analysis for S2 (Fig. 4). For simulation, why don't you show the phase analysis for the same S2? why did you choose SW2 instead?
**R: well, for two main reasons. Firstly, because we fitted each tidal component separately. Hence, mixing the phases of all tidal components contributing to S2 would be misleading. Secondly, because after realizing that SW2 strongly dominates the behavior of S2 during the fall, we mainly focused our study on this component (in the simulations).**

16. 9/25: I agree with the authors that GW-tide interaction requires a thorough analysis, given that not only the simulation, but also your observations contain only a certain part of the GW spectrum, and different parts of the GW spectrum can interact differently with tides. The interaction of other parts of the GW spectrum with S2 cannot be estimated here and may also influence the S2 variability. Maybe you should add one sentence here to clarify that fact.
**R: thanks. We have added a sentence on this matter. Please see in the revised manuscript: page 9, line 35, and page 10, line 1.**

17. 9/27: "the GW activity" → "the long-wave GW activity"
**R: thanks for this comment. We have applied this change. Please see in the revised manuscript: page 10, line 5.**

18. 9/30: Would you conclude that long-wave GWs suppress the migrating semidiurnal tide SW2 amplitude?
**R: we think that the GW-tide interaction could be one of the causes of the fall decrease of SW2. However, we still don't have sufficient evidence to firmly conclude this, and that is why we are only suggesting that GWs could be involved.**

[revised manuscript text omitted]